# Fully Dynamic Algorithm
# for Constrained Submodular Optimization

**Silvio Lattanzi**\*
Google Research
silviol@google.com

**Slobodan Mitrović**\*
MIT
slobo@mit.edu

**Ashkan Norouzi-Fard**\*
Google Research
ashkannorouzi@google.com

**Jakub Tarnawski**\*
Microsoft Research
jatarnaw@microsoft.com

**Morteza Zadimoghaddam**\*
Google Research
zadim@google.com

## Abstract

The task of maximizing a monotone submodular function under a cardinality constraint is at the core of many machine learning and data mining applications, including data summarization, sparse regression and coverage problems. We study this classic problem in the fully dynamic setting, where elements can be both inserted and removed. Our main result is a randomized algorithm that maintains an efficient data structure with a poly-logarithmic amortized update time and yields a $(1/2 - \epsilon)$-approximate solution. We complement our theoretical analysis with an empirical study of the performance of our algorithm.

## 1   Introduction

Thanks to the ubiquitous nature of "diminishing returns" functions, submodular optimization has established itself as a central topic in machine learning, with a myriad of applications ranging from active learning [GK11] to sparse reconstruction [Bac10, DDK12, DK11], video analysis [ZJCP14] and data summarization [BIRB15]. In this field, the problem of maximizing a monotone submodular function under a cardinality constraint is perhaps the most central. Despite its generality, the problem can be (approximately) solved using a simple and efficient greedy algorithm [NWF78].

However, this classic algorithm is inefficient when applied on modern large datasets. To overcome this limitation, in recent years there has been much interest in designing efficient streaming [BMKK14, CK14, BFS15, FKK18, NTM+18] and distributed algorithms [MZ15, MKBK15, BENW16, ENV19] for submodular maximization.

Although those algorithms have found numerous applications, they are not well-suited for the common applications where data is highly dynamic. In fact, real-world systems often need to handle evolving datasets, where elements are added and deleted continuously. For example, in a recent study [DJR12], Dey et al. crawled two snapshots of 1.4 million New York City Facebook users several months apart and reported that 52% of them had changed their profile privacy settings significantly during this period. Similarly, Snapchat processes several million picture uploads and deletions daily; Twitter processes several million tweet uploads and deletions daily. As one must still be able to run basic machine learning tasks, such as sparse recovery or data summarization, in such highly dynamic settings, we need fully *dynamic algorithms*: ones able to *efficiently* handle a stream containing not only insertions, but also an arbitrary number of deletions, with small processing time per update.

---

The general dynamic setting is classic and a staple of algorithm design, with many applications in machine learning systems. However, for many problems it is notoriously difficult to obtain efficient algorithms in this model. In the case of submodular maximization, algorithms have been proposed for the specialized settings of sliding windows [CNZ16, ELVZ17] and robustness [MBN+17, KZK18]. However, as we discuss below, these solutions cannot handle the full generality of the described real-world scenarios.

**Our contribution.** In this paper we design an efficient fully dynamic algorithm for submodular maximization under a cardinality constraint. Our algorithm:

- takes as input a sequence of arbitrarily interleaved insertions and deletions,

- after each such update, it continuously maintains a solution whose value is in expectation at least $(1/2 - \epsilon)$ times the optimum of the underlying dataset at the current time,

- has amortized time per update that is poly-logarithmic in the length of the stream.

This result settles the status of submodular maximization as *tractable* in the dynamic setting. We also empirically validate the efficiency of our algorithm in several applications.

**Related work.** The question of computing a concise summary of a stream of $n$ data points on the fly was first addressed by *streaming* algorithms. This line of work focuses on using small space, independent of (or only poly-logarithmically dependent on) $n$. The SIEVESTREAMING algorithm [BMKK14] achieves a $(1/2 - \epsilon)$-approximation in this model, which is tight [FNFSZ20]. The main thresholding idea of SIEVESTREAMING has had a large influence on recent submodular works, including ours. However, streaming algorithms do not support deletions. In fact, the low-memory requirement is fundamentally at odds with the dynamic setting, as any approximation algorithm for the latter must store all stream elements.[2] A natural idea is to adapt streaming algorithms to deletions by storing the stream and recomputing the solution when it loses elements. However, this takes $\Omega(n)$ time per deletion, and is also shown to be inefficient in our experimental evaluations.

A notable related problem is that of maintaining a summary that focuses only on recent data (e.g., the most recent one million data points). This task is captured by the *sliding window model*. In particular, [CNZ16, ELVZ17] give algorithms that optimize a monotone submodular function under the additional constraint that only the last $W$ elements of the stream can be part of the solution. Unfortunately this setting, while crucial for the data freshness objective, is unrealistic for real-world dynamic systems, where it is impossible to assume that data points are deleted in such structured order. In particular, emerging privacy concerns and data protection regulations require data processing platforms to respond rapidly to users' data removal requests. This means that the arrival and removal of data points follows an arbitrary and non-homogeneous pattern.

Another important task is that of generating a summary that is robust to a specific number $D$ of adversarial deletions. This setting is the inspiration for the *two-stage deletion-robust model*. In the first stage, elements are inserted, and the algorithm must retain an intermediate summary of limited size. In the second stage, an adversary removes a set of up to $D$ items. The algorithm then needs to find a final solution from the intermediate summary while excluding the removed items. Despite the generality of the deleted items being arbitrary, this framework assumes that all deletions occur after all items have been introduced to the system, which is often unrealistic and incompatible with privacy objectives. Furthermore, in the known algorithms for this setting [MBN+17, KZK18], the time needed to compute a single solution depends linearly on $D$, which could be as large as the size $n$ of the entire dataset. Therefore a straightforward use of these methods in fully dynamic settings would result in $\Omega(n)$ per-update time, which is prohibitively expensive.

Finally, a closely related area is that of low-adaptivity complexity. In particular, [FMZ19] is closely related to our work; we build upon the batch insertion idea of the Threshold Sampling algorithm introduced there.

## 2 Preliminaries

We consider a (potentially large) collection $V$ of items, also called the *ground set*. We study the problem of maximizing a *non-negative monotone submodular function* $f : 2^V \to \mathbb{R}_{\geq 0}$. Given two sets $X, Y \subseteq V$, the *marginal gain* of $X$ with respect to $Y$ is defined as

$$f(X \mid Y) = f(X \cup Y) - f(Y),$$

which quantifies the increase in value when adding $X$ to $Y$. We say that $f$ is *monotone* if for any element $e \in V$ and any set $Y \subseteq V$ it holds that $f(e \mid Y) \geq 0$. The function $f$ is *submodular* if for any two sets $X$ and $Y$ such that $X \subseteq Y \subseteq V$ and any element $e \in V \setminus Y$ we have

$$f(e \mid X) \geq f(e \mid Y).$$

Throughout the paper, we assume that $f$ is *normalized*, i.e., $f(\emptyset) = 0$. We also assume that $f$ is given in terms of a value oracle that computes $f(S)$ for given $S \subseteq V$. As usual in the field, when we talk about running time, we are counting the number of oracle calls/queries, each of which we treat as a unit operation. The number of non-oracle-call operations we perform is within a polylog factor of the number of oracle calls.

**Submodularity under a cardinality constraint.** The problem of maximizing a function $f$ under a *cardinality constraint* $k$ is defined as selecting a set $S \subseteq V$ with $|S| \leq k$ so as to maximize $f(S)$. We will use OPT to refer to such a maximum value of $f$.

**Notation for dynamic streams.** Consider a stream of insertions and deletions. Denote by $V_i$ the set of all elements that have been inserted and not deleted up to the $i$-th operation. Let $\mathcal{O}_i$ be an optimum solution for $V_i$; denote $\mathrm{OPT}_i = f(\mathcal{O}_i)$.

In our dynamic algorithm we are interested in updating our data structure efficiently. We say that an algorithm has amortized update time $t$ if its total running time to process a worst-case sequence of $n$ insertions and deletions is in expectation at most $nt$.

## 3 Overview of our approach and intuitive analysis

In this section we provide an overview of the main techniques and ideas used in our algorithm. To that end we skip some details of the algorithm and present the arguments intuitively, while formal arguments are provided in Section 4. We start by noting that previous approaches either do not support deletions altogether, or support only a limited (and small) number of deletions (with linear running time per deletion) and so they do not capture many real-world scenarios. In this work, we overcome this barrier by designing a novel fully dynamic data structure that has only poly-logarithmic amortized update time.

We start with a few useful observations. For a moment, ignore the values of elements in the ground set $V$. Consider a set $X$ of $k$ elements sampled uniformly at *random* from $V$.[3] The set $X$ is very robust against deletions (which, as a reminder, we asssume to be chosen *independently* of the choice of $X$). Namely, in order to delete an $\epsilon$-fraction of the elements in $X$, one needs (in expectation) to delete an $\epsilon$-fraction of the elements in $V$. This property suggests the following fast algorithm, that we refer to by ALG-SIMPLE, for maintaining a set of at most $k$ elements: sample $k$ elements uniformly at random from the ground set, and call that set $X$; after an $\epsilon$-fraction of the elements in $X$ is deleted, sample another $X$ from scratch. The current set $X$ represents an output after an update. ALG-SIMPLE has expected running time $O(1/\epsilon)$ per deletion, and can also be extended to support insertions in $\mathrm{polylog}(n)$ time. The main issue with this approach is the lack of guarantees on the quality of the output solution after an update, i.e., the approach is oblivious to the values of the elements in $V$. For instance, the ground set might contain many *useless* elements, hence selecting $k$ of them uniformly at random would not lead to a set of high utility. The main idea in our paper is to partition the ground set into groups (that we call *buckets*) so that applying ALG-SIMPLE within each bucket outputs a robust set of high utility. Moreover, the union of these sampled elements provides close to optimal utility.

Our data structure, which we refer to by $A$, divides the elements into $T = \log n$ *levels*, with each level subdivided into $R = \log k$ buckets. Informally speaking, each bucket is designed in such a way that selecting elements from it by ALG-SIMPLE results in sets that are both robust and high-quality. Our algorithm maintains a set $S$ that represents the output solution at every point; it is constructed by applying ALG-SIMPLE over distinct buckets. Different buckets might contribute different numbers of elements to $S$.

The structure of each level of $A$ is essentially the same. The main difference is that different levels maintain different numbers of elements, i.e., level $\ell$ maintains $O(\frac{n}{2^\ell} \cdot \text{polylog}(n))$ many elements. Intuitively, and informally, levels with small $\ell$ are recomputed/changed only after many updates, while levels with large $\ell$, such as $\ell = T$, are sensitive to updates and recomputed more frequently. In particular, if we insert an extremely valuable element, then the level $\ell = T$ will guarantee that this newly added valuable element will appear in $S$. We now discuss the structure of $A$ in more detail.

We use $A_{i,\ell}$ to refer to the $i$-th bucket in level $\ell$. Each level is associated with a maximum bucket-size, with level 0 corresponding to the largest bucket-sizes. More precisely, we will maintain the invariant

$$|A_{i,\ell}| \leq \frac{n}{2^\ell} \cdot \text{polylog}(n)$$

for all $1 \leq i \leq R$. Organizing levels to correspond to exponentially decreasing bucket-sizes is one of the main ingredients that enables us to obtain a poly-logarithmic update time.

Buckets within each level are ordered so as to contain elements of exponentially decreasing marginal values with respect to the elements chosen so far. To illustrate this partitioning, consider the first bucket of level 0. Let $S$ be the set of elements representing our (partial) output so far; initially, $S = \emptyset$. Then, we define

$$A_{1,0} = \{e \in V \mid \tau_1 \leq f(e \mid S) \leq \tau_0\},$$

where $\tau_i \approx (1-\epsilon)^i \text{ OPT}$.[4] It is clear that the construction of $A_{1,0}$ takes $\widetilde{O}(n)$ time. After constructing $A_{1,0}$, our goal is to augment $S$ by some of the elements from $A_{1,0}$ so that the marginal gain of each element added to $S$ is in expectation at least $\tau_1$. After augmenting $S$, we also refine $A_{1,0}$. This is achieved by repeatedly performing the following steps:
From $A_{1,0}$ we randomly select a subset (of size at most $k - |S|$) of elements whose average marginal gain with respect to to $S$ is at least $\tau_1$. In Appendix C we explain how to obtain such a set efficiently. Then we add this set to $S$. Now, refine $A_{1,1}$ by removing from it all elements whose marginal gain with respect to $S$ is less than $\tau_1$. If $|A_{1,0}| \geq n/2$ and $|S| < k$, we repeat these steps.

Let us now analyze the robustness of $S \cap A_{1,0}$. The way we selected the elements added to $S$ enables us to perform a similar reasoning to the one we performed to analyze the robustness of ALG-SIMPLE. Namely, when an element $e \in A_{1,0}$ is added to $S$, it is always chosen *uniformly at random* from $A_{1,0}$. Also, the process of adding elements to $S$ from $A_{1,0}$ is repeated while $|A_{1,0}| \geq n/2$. In other words, $e$ is chosen from a large pool of elements, much larger than $k$. Hence, an adversary has to remove many elements, $\varepsilon|A_{i,\ell}| \geq \varepsilon n/2$ in expectation, to remove an $\varepsilon$-fraction of elements added from $A_{i,\ell}$ to $S$. Combining this observation with the fact that the construction of $A_{1,0}$ takes $\widetilde{O}(n)$ time is key to obtaining to the desired update time[5].

Note that so far we have assumed that a good solution can be constructed looking only at elements with marginal value larger than $\tau_1$. Unfortunately this is not always the case and so we need to extend our construction. To construct the remaining buckets $A_{i,\ell}$, we proceed in the same fashion as for $A_{1,0}$ in the increasing order of $i$. The only difference is that we consider decreasing thresholds:

$$A_{i,\ell} = \{e \in V \mid \tau_i \leq f(e \mid S) \leq \tau_{i-1}\},$$

where $S$ is always the set of elements chosen so far. Once all the buckets in level 0 are processed, we proceed to level 1. The main difference between different layers is that for level $\ell$ we iterate while $|A_{i,\ell}| \geq n/2^\ell$ and $|S| < k$. So, in every level we explore more of our ground set. Importantly, we can show that on every level we consider a ground set that decreases in size significantly.

At first, it might be surprising that from bucket to bucket of level $\ell$ we consider elements in decreasing order of their marginal gain, and then in level $\ell + 1$ we again begin by considering elements of the

largest gain. Perhaps it would be more natural to first exhaust all the elements of the largest marginal gain, and only then consider those of lower gain. However, we remark that the smallest value of $\tau_i$ that we consider is at least $\Theta(\text{OPT}/k)$. Hence, selecting for $S$ any $k$ elements whose marginal contribution is at least $\tau_i$ already leads to a good approximation.

**Handling Deletions.** Assume that an adversary deletes an element $e$. If $e \notin S$, we remove $e$ only from the buckets it belongs to, without any extra recomputation. If $e \in S$, let $A_{i,\ell}$ be the bucket from which $e$ is added to $S$. To update $S$, we reconstruct $A$ from $A_{1,\ell}$. We now informally bound the running time needed for this reconstruction. The probability that an element from $A_{1,\ell}$ belongs to $S$ is $\frac{t}{n/2^\ell}$, where $t$ is the number of elements selected to $S$ from $A_{i,\ell}$. Saying it differently, an adversary has to (in expectation) remove $\frac{n/2^\ell}{t}$ elements from $A_{i,\ell}$ before it removes an element from $S \cap A_{i,\ell}$. Moreover the running time of a reconstruction of $A_{1,\ell}$ is $\widetilde{O}(n/2^\ell)$. Putting these two together, we get that expected running time of reconstruction per deletion is $O(t) \cdot \text{polylog}(n)$. To reduce the update time to $\text{polylog}\, n$, we reconstruct $S$ only if, since its last reconstruction, its value has dropped by a factor $\varepsilon$. Since the elements in $A_{i,\ell}$ have similar marginal gain, an adversary would need to remove roughly $\varepsilon t$ elements from $S \cap A_{i,\ell}$ to invoke a recomputation of $A_{1,\ell}$, leading to an amortized update time of $\text{polylog}(n)$. Unfortunately, formalizing this intuition is somewhat subtle, as elements are removed from multiple buckets and each removal decreases the value of $S$.

**Handling Insertions.** Along with $A$, we maintain buffer sets $B_1, \ldots, B_T$. Roughly speaking, our algorithm postpones processing insertions into level $\ell$ until there are $n/2^\ell$ many of them; this enables us to obtain efficient amortized update time of the structure on level $\ell$. The buffer set $B_\ell$ is used to store these insertions until they are processed.

More precisely, when an element is inserted, the algorithm adds it to all the sets $B_\ell$. When, for any $\ell$, the size of $B_\ell$ becomes $\frac{n}{2^\ell}$, we add the elements of $B_\ell$ to $A_{1,\ell}$, reconstruct the data structure from the $\ell$-th level, and also empty $B_\ell$. This approach handles insertions *lazily*. Notice that lazy updates should be done carefully, since if the newly inserted element has very high utility, we need to add it to the solution immediately. During the execution of the algorithm, $B_\ell$ essentially represents those elements that we have not considered in the construction of buckets in $A_{i,\ell}$ for $0 \leq \ell \leq T$. The property that the running time of constructing $A_{i,\ell}$ is $\widetilde{O}(\frac{n}{2^\ell})$ implies that the amortized running time per insertion is also $\text{polylog}(n)$. Also observe that we add $B_T$ to $A_{1,T}$ after any element is inserted, which enables us to maintain a good approximate solution at all times. In particular, if an element $e$ of very large marginal gain given $S$ is inserted, e.g., $f(e \mid S) > \text{OPT}/2$, then it will be processed via $B_T$ and added to $S$. In general, if there are $2^j$ inserted elements that collectively have very large gain given $S$, then they will be processed via $B_{T-j}$ and potentially used to update $S$.

## 4 The algorithm

We are now ready to describe our algorithm. For the sake of simplicity, we present an algorithm that is parametrized by $\gamma$: a guess for the value OPT. Moreover we assume that we know the maximum number of elements available at any given time ($\max_{1 \leq t \leq m} |V_t|$), which is upper-bounded by $n$. Later we show how to remove these assumptions.

Our algorithm maintains a data structure that uses three families of element sets: $A$ and $S$ indexed by pairs $(i, \ell)$ and $B$ indexed by $\ell$. For an integer $R$ that we will set later, the algorithm also maintains a sequence of thresholds $\tau_0 > \ldots > \tau_R$ (indexed by $i$), where we think that $\tau_0 \approx \gamma$ and $\tau_R \approx \gamma/(2k)$. We use $S_{j,\ell}$ to refer to the elements chosen to $S$ from bucket $j$ of level $\ell$. Let $S_{\text{pred}(i,\ell)}$ be the following union of sets:

$$S_{\text{pred}(i,\ell)} \stackrel{\text{def}}{=} \bigcup_{1 \leq j \leq R, 0 \leq r < \ell} S_{j,r} \cup \bigcup_{1 \leq j \leq i} S_{j,\ell}.$$

In words, a set $S_{\text{pred}(i,\ell)}$, where "pred" refers to "predecessors", defines the subset of elements of $S$ chosen from the buckets that precede bucket $i$ of level $\ell$, including that bucket itself. At level $\ell$ and for index $i$, we define $A_{i,\ell}$ to be the set of items with marginal value with respect to the set $S_{\text{pred}(i,\ell)}$ in the range $[\tau_i, \tau_{i-1}]$. While $A_{i,\ell}$ has at least $2^{T-\ell}$ items, we use a procedure called

PEELING[6] to select a random subset of $A_{i,\ell}$ to be included into the solution set $S_{i,\ell}$. This can be done in multiple iterations; each time, a randomly chosen batch of items will be inserted into $S_{i,\ell}$. This batch insertion logic is named BUCKET-CONSTRUCT and summarized as Algorithm 2. The solution that our algorithm returns is $S_{\mathrm{pred}(R,T)}$, i.e., the union of all sets $S_{i,\ell}$, and we denote by $\mathrm{Sol}_t$ this set after the $t$-th operation.

In order to implement our algorithm efficiently, we need to be able to select a high-quality random subset of $A_{i,\ell}$ quickly. Our data structure enables us to do this using the PEELING procedure from [FMZ19] (whose full description and a precise statement and proofs of its guarantees are provided in Appendix C).[7] This procedure takes as input a set $N$ and identifies a number $t$ and selects a set $S$ of size $t$ uniformly at random such that: i) the average contribution of each element in $S$ is almost $\tau$, ii) a large fraction of elements in $N$ have contribution less than $\tau$, conditioned on adding $S$ to the solution, iii) it uses only a logarithmic number of oracle queries.

To maintain the above batch insertion logic with every insertion, the algorithm may need to recompute many of the $A$-sets, which blows up the update time. To get around this problem, we introduce buffer sets $B_\ell$ for each level $0 \leq \ell \leq T$. Each buffer set $B_\ell$ has a capacity of at most $2^{T-\ell} - 1$ items. When a new item $x$ arrives, instead of recomputing all $A$-sets, we insert $x$ into all buffer sets. If some buffer sets exceed their capacity, we pick the first one (with the smallest $\ell^*$) and reconstruct all sets in levels beginning from $\ell^*$. We call this reconstruction process LEVEL-CONSTRUCT. It is presented as Algorithm 5. The insertion process in summarized as Algorithm 3.

When deleting an element $x$, our data structure is not affected if the deleted item $x$ does not belong to any set $S_{i,\ell}$. But if it is deleted from some $S_{i,\ell}$, we need to recompute the data structure starting from $S_{i,\ell}$. To optimize the update time, we perform this update operation in a lazy manner as well. We recompute only if an $\varepsilon$-fraction of items in $S_{i,\ell}$ have been deleted since the last time it was constructed. To simplify the algorithm, we reconstruct the entire level $\ell$ and also the next levels $\ell + 1, \dots$ in this case. The deletion logic is summarized as Algorithm 4.

We initialize all sets as empty. The sequence of thresholds $\tau$ is set up as a geometric series parametrized by a constant $\epsilon_1 > 0$.

# 5   Analysis of the algorithm

We now state two technical theorems, and in Appendix C.1 we show how to combine them in the main result. Here $\epsilon_1, \epsilon_p > 0$ are parameters of our algorithm; they affect both approximation ratio and oracle complexity. Intuitively, they should be thought of as small constants. (As a reminder, our approach consists of five methods INITIALIZATION, BUCKET-CONSTRUCT, INSERTION, DELETION and LEVEL-CONSTRUCT that are given as Algorithm 1 through Algorithm 5.)

**Theorem 5.1** *Let $Sol_i$ be the solution of our algorithm and $\mathrm{OPT}_i$ be the optimal solution after $i$ updates. Moreover, assume that $\gamma$ in Algorithm 1 is such that $(1 + \epsilon_p)\,\mathrm{OPT}_i \geq \gamma \geq \mathrm{OPT}_i$. Then for any $1 \leq i \leq n$ we have $\mathbb{E}[f(Sol_i)] \geq (1 - \epsilon_p - \epsilon(1 + \epsilon_1))\frac{\mathrm{OPT}_i}{2}$.*

**Theorem 5.2** *The amortized expected number of oracle queries per update is $O\left(\frac{R^5 \log^2(n)}{\epsilon_p^2 \cdot \varepsilon}\right)$, where $R$ equals $\log_{1+\epsilon_1}(2k)$ (see Algorithm 1).*

Theorems 5.1 and 5.2 are proved in Appendices A and B, respectively. Furthermore, we combine these ingredients with certain well-known techniques to achieve the following result. Its proof is provided in Appendix C.1.

**Theorem 5.3** *Our algorithm maintains a $(1 - 2\epsilon_p - \epsilon(1 + \epsilon_1))/2$-approximate solution after each operation. The amortized expected number of oracle queries per update of this algorithm is $O\left(\frac{\log_{1+\epsilon_1}^6(k) \log^2(n)}{\epsilon_p^4 \cdot \varepsilon}\right)$.*

**Algorithm 1** INITIALIZATION

1: $R \leftarrow \log_{1+\epsilon_1}(2k)$
2: $\tau_i \leftarrow \gamma(1+\epsilon_1)^{-i}$ $\forall 0 \le i \le R$
3: $T \leftarrow \log n$
4: $A_{i,\ell} \leftarrow \emptyset$ $\forall 1 \le i \le R$ $0 \le \ell \le T$
5: $S_{i,\ell} \leftarrow \emptyset$ $\forall 1 \le i \le R$ $0 \le \ell \le T$
6: $B_\ell \leftarrow \emptyset$ $\forall 0 \le \ell \le T$

---

**Algorithm 2** BUCKET-CONSTRUCT$(i, \ell)$

1: **repeat**
2: $A_{i,\ell} = \{e \in A_{i,\ell} \mid \tau_i \le f(e \mid S_{\text{pred}(i,\ell)}) \le \tau_{i-1}\}$
3: **if** $|A_{i,\ell}| \ge 2^{T-\ell}$ and $|S_{\text{pred}(R,T)}| < k$ **then**
4: $S_{i,\ell} \leftarrow S_{i,\ell} \cup \text{PEELING}(A_{i,\ell}, \tau_i, f')$
5: **end if**
6: **until** $|A_{i,\ell}| < 2^{T-\ell}$ or $|S_{\text{pred}(R,T)}| \ge k$

---

**Algorithm 3** INSERTION$(e)$

1: $B_\ell \leftarrow B_\ell \cup \{e\}$ $\forall 0 \le \ell \le T$
2: $V \leftarrow V \cup \{e\}$
3: **if** there exists an index $\ell$ such that $|B_\ell| \ge 2^{T-\ell}$ **then**
4: Let $\ell^\star$ be the smallest such index
5: $S_{i',\ell'} \leftarrow \emptyset$ $\forall \ell^\star \le \ell' \le T$ $\forall 1 \le i' \le R$
6: $B_\ell \leftarrow \emptyset$ $\forall \ell^\star \le \ell' \le T$
7: LEVEL-CONSTRUCT$(\ell^\star)$
8: **end if**

---

**Algorithm 4** DELETION$(e)$

1: $A_{i,\ell} \leftarrow A_{i,\ell} \setminus \{e\}$ $\forall 1 \le i \le R$ $0 \le \ell \le T$
2: $B_\ell \leftarrow B_\ell \setminus \{e\}$ $\forall 0 \le \ell \le T$
3: $V \leftarrow V \setminus \{e\}$
4: **if** $e \in S_{\text{pred}(R,T)}$ **then**
5: Let $S_{i,\ell}$ be the set containing $e$
6: Remove $e$ from $S_{i,\ell}$
7: **if** the size of $S_{i,\ell}$ has reduced by $\varepsilon$ fraction since it was constructed **then**
8: $S_{i',\ell'} \leftarrow \emptyset$ $\forall \ell \le \ell' \le T$ $\forall 0 \le i' \le R$
9: LEVEL-CONSTRUCT$(\ell)$
10: **end if**
11: **end if**

---

**Algorithm 5** LEVEL-CONSTRUCT$(\ell)$

1: $B_\ell \leftarrow \emptyset$
2: **for** $i \leftarrow 1 \ldots R$ **do**
3: **if** $\ell > 0$ **then**
4: $A_{i,\ell} \leftarrow B_{\ell-1} \cup \bigcup_{j=0}^{R} A_{j,\ell-1}$
5: **else**
6: $A_{i,\ell} \leftarrow V$
7: **end if**
8: BUCKET-CONSTRUCT$(i, \ell)$
9: **end for**
10: **if** $|S_{\text{pred}(R,T)}| \ge k$ **then**
11: $A_{i,\ell'} \leftarrow \emptyset$ $\forall \ell < \ell' \le T$ $\forall 1 \le i \le R$
12: **end if**
13: **if** $\ell < T$ and $|S_{\text{pred}(R,T)}| < k$ **then**
14: LEVEL-CONSTRUCT$(\ell + 1)$
15: **end if**

## 6 Empirical evaluation

In this section we empirically evaluate our algorithm. We perform experiments using a slightly simplified variant of our algorithm; see Appendix E for more details. We note that this variant also maintains an almost $1/2$-approximate solution after each operation; it differs in the bound on the expected number of oracle queries per update that we can obtain, which is $\tilde{O}(k)$. A proof of these guarantees can also be found in Appendix E.

The code of our implementations can be found at `https://github.com/google-research/google-research/tree/master/fully_dynamic_submodular_maximization`. All experiments in this paper are run on commodity hardware.

We focus on the number of oracle calls performed during the computation and on the quality of returned solutions. More specifically, we perform a sequence of insertions and removals of elements, and after each operation $i$ we output a high-value set $S_i$ of cardinality at most $k$. For a given sequence of $n$ operations, we plot:

- **Total number of oracle calls** our algorithm performs for each of the $n$ operations.

- **Quality** of the average output set, i.e., $\sum_{i=1}^{n} f(S_i)/n$.

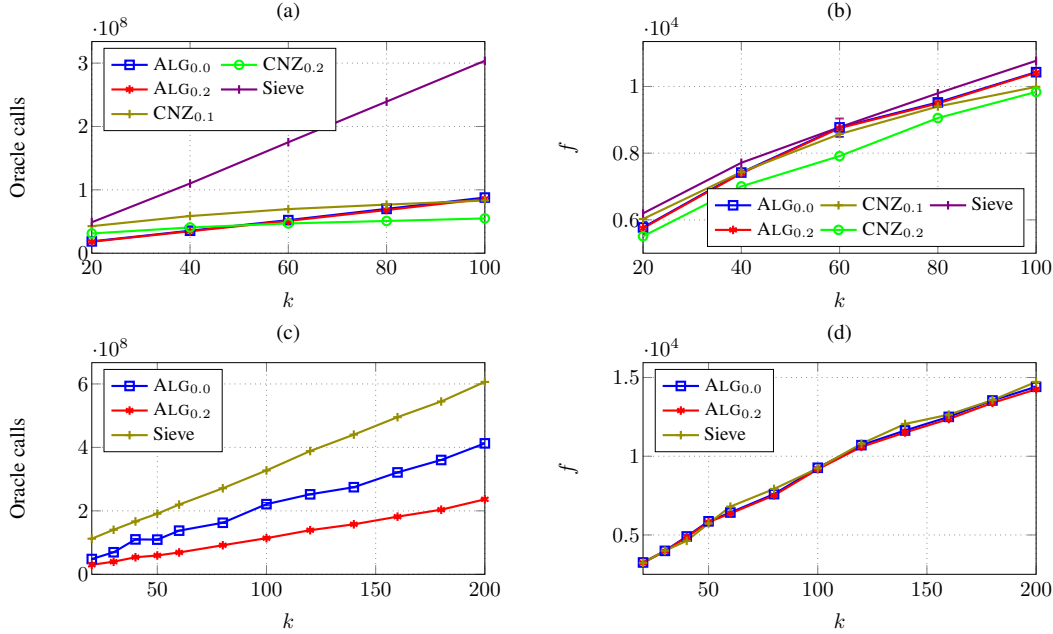

Figure 1: The plots in this figure are obtained for $f$ being the graph coverage function. Plots (a) and (b) show the results on the Enron dataset. We fix an arbitrary order of the Enron email addresses and process them sequentially over windows of size $30,000$. We first insert all elements, and then delete them in the same order. Plots (c) and (d) depict the results for the ego-Twitter dataset. In this experiment the insertions are performed in a random order, while deletions are performed starting from highest-degree nodes.

**Dominating sets.** In our evaluation we use the dominating set objective function. Namely, given a graph $G = (V, E)$, for a subset of nodes $Z \subseteq V$ we define $f(Z) = |N(Z) \cup Z|$, where $N(Z)$ is the node-neighborhood of $Z$. This function is monotone and submodular.

**Datasets and their processing.** We perform evaluations on the Enron ($|V| = 36,692, |E| = 183,831$), the ego-Twitter ($|V| = 81,306, |E| = 1,768,149$), and the Pokec ($|V| = 1,632,803, |E| = 30,622,564$) graph datasets from SNAP Large Networks Data Collection [LK15].

We run two types of experiments on the abovementioned datasets.

1. We consider **a sliding window** of size $\ell$ over an arbitrary order of the nodes of the graph. When the window reaches a node, we add that node to the stream. Similarly, after $\ell$ insertions, i.e., when a node leaves the window, we delete it. This provides us with a stream of interspersed insertions and deletions. Moreover, setting $\ell$ to the number of nodes in the graph is equivalent to inserting all the nodes in an arbitrary order and then deleting them in the same order.

2. We insert all the nodes of the graph in **arbitrary order**. Afterward, we delete them node-by-node by choosing a node in the current solution that has the largest neighborhood. Intuitively, we delete the elements that contribute the most to the optimum solution; this potentially results in many changes to $\mathrm{Sol}_i$. We observe that even for this stream, our algorithm is efficient and makes a small number of oracle calls on average.

Due to space constraints, we present the results of only two experiments, one for each of the types. For the first type, we present the results on the Enron dataset for a window of size $\ell = 30,000$. For the second type, we present the results on the ego-Twitter dataset. Further results on other datasets and different values of $\ell$ are included in Appendix D.

**The baselines.** We consider the performance of our algorithm for $\epsilon = 0.0$ and $\epsilon = 0.2$, and denote those versions by $\mathrm{ALG}_{0.0}$ and $\mathrm{ALG}_{0.2}$, respectively. Recall that in our algorithm, if an $\epsilon$-fraction of elements is deleted from the solution on some level, we reconstruct the solution beginning from that

level. We cannot compare against the true optimum or the greedy solution, as computing them is intractable for data of this size. We compare our approach with the following baselines:

1. The algorithms of [CNZ16] and [ELVZ17] (developed concurrently and very similar). This method is designed for the sliding window setting and can only be used if elements are deleted in the same order as they were inserted. It is parametrized by $\varepsilon$ and we consider values of $\varepsilon = 0.1$ and $\varepsilon = 0.2$, and use $\text{CNZ}_{0.1}$ and $\text{CNZ}_{0.2}$ to denote these two variants.

2. SIEVESTREAMING [BMKK14], which is a streaming algorithm that only supports inserting elements. For any insertion, we simply have SIEVESTREAMING insert the element. For any deletion that deletes an element in the solution of SIEVESTREAMING, we restart SIEVESTREAMING on the set of currently available elements.[8]

3. RND algorithm, which maintains a uniformly random subset of $k$ elements. RND outputs solutions of significantly lower quality than other baselines, so due to space constraints we report its objective value results only in the appendix.

**Results.** The results of our evaluation are presented in Fig. 1. As shown in plots (b) and (d), our approach (even for different values of $\varepsilon$) is qualitatively almost the same as SIEVESTREAMING. However, compared to SIEVESTREAMING, our approach has a smoother increase in the number of oracle calls with respect to the increase in $k$. As a result, starting from small values of $k$, e.g., $k = 40$, our approach $\text{ALG}_{0.2}$ requires at least $2\times$ fewer oracle calls than SIEVESTREAMING to output sets of the same quality for both Enron and ego-Twitter. The behavior of our algorithm for $\varepsilon = 0.0$ is closest to SIEVESTREAMING in the sense that, as soon as a deletion from the current solution occurs, it performs a recomputation (see Line 7 of DELETION). For larger $\varepsilon$ our approach performs a recomputation only after a number of deletions from the current solution. As a result, for $\varepsilon = 0.2$, on some datasets our approach requires almost $3\times$ fewer oracle calls to obtain a solution of the same quality as SIEVESTREAMING (see Fig. 1(c) and (d)).

Compared to $\text{CNZ}_{0.1}$ and $\text{CNZ}_{0.2}$ in the context of sliding-window experiments (plots (a) and (b) in Fig. 1), our approach shows very similar performance in both quality and the number of oracle calls. $\text{CNZ}_{0.2}$ is somewhat faster than our approach (plot (a)), but it also reports a lower-quality solution (plot (b)). We point out that CNZ fundamentally requires that insertions and deletions are performed in the same order. Hence, we could not run CNZ for plots (c) and (d), where the experiment does not have that special structure. Since our approach is randomized, we repeat each of the experiments 5 times using fresh randomness; plots show the mean values. The standard deviation of reported values for $\text{ALG}_{0.0}$ and $\text{ALG}_{0.2}$, less than $5\%$, is plotted in Fig. 8.

## 7   Conclusion and future work

We present the first efficient algorithm for cardinality-constrained dynamic submodular maximization, with only poly-logarithmic amortized update time. We complement our theoretical results with an extensive experimental analysis showing the practical performance of our solution. Our algorithm achieves an almost $1/2$-approximation. This approximation ratio is tight in the (low-memory) streaming setting [FNFSZ20], but not necessarily in the dynamic setting; a natural question is whether it can be improved, even for insertion-only streams. Another compelling direction for future work is to extend the current result to more general constraints such as matroids.

## Broader impact

This work does not present any foreseeable societal consequence.

## Acknowledgments and Disclosure of Funding

Slobodan Mitrović was supported by the Swiss NSF grant No. P400P2_191122/1, MIT-IBM Watson AI Lab and Research Collaboration Agreement No. W1771646, and FinTech@CSAIL.

## Footnotes

[2]If even one element is not stored by an algorithm, an adversary could delete all other elements, bringing the approximation ratio down to 0.

[3] Hence, each element from $V$ is sampled with probability $k/n$.

[4]As a reminder, OPT denotes the maximum value of $f(S)$ over all $S \subseteq V$ such that $|S| \leq k$.

[5]Note that actually achieving the desired running time without any assumption requires further adjustments to the algorithm and more involved techniques that we introduce in further sections.

[6]Algorithm PEELING is an implementation of the ideas behind ALG-SIMPLE described in Section 3.

[7]We invoke PEELING on the function $f'(e) = f(e \mid S_{\mathrm{pred}(i,\ell)})$, which is monotone submodular.

[8]Like our algorithm, SIEVESTREAMING operates parallel copies of the algorithm for different guesses of OPT. We restart only those copies whose solution contains the removed element.

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
