[Supplementary Material]

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

[9]Recall that $T = \log n$ (see Line 3 of INITIALIZATION.)

[10]This might not happen immediately after the insertion.

[11]Recall that, in case $\ell = 0$, by $S_{\mathrm{pred}(r,\ell-1)}$ we denote $S_{\mathrm{pred}(r-1,T)}$. Moreover, we let $S_{\mathrm{pred}(-1,T)} = \emptyset$.

[12]Here we do not provide a formal proof, since in the previous sections we present a more efficient algorithm that obtains better running times for both insertions and deletions.

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

# A  Query complexity of our algorithm

We now state two invariants that are maintained by our algorithms. We will then use these invariants to analyze the oracle-query complexity of our algorithms.

**Invariant 1** *For any $\ell$, it holds that $|A_\ell| \leq (R+1) \cdot 2^{T-\ell}$.[9] Here we overload notation $A_\ell$ to denote $\cup_{i=1}^{R} A_{i,\ell}$.*

**Invariant 2** *For any $\ell$, it holds that $|B_\ell| \leq 2^{T-\ell}$.*

**Lemma A.1** *If Invariants 1 and 2 hold before invoking* LEVEL-CONSTRUCT *(Algorithm 5), then they also hold after executing Line 13 of* LEVEL-CONSTRUCT. *Consequently, the invariants hold after* LEVEL-CONSTRUCT *terminates.*

*Proof.* First, notice that the invocation of LEVEL-CONSTRUCT$(\ell')$ never adds new elements to any $B_\ell$, hence the claim holds for Invariant 2. Now we prove the claim for Invariant 1.

Notice that LEVEL-CONSTRUCT$(\ell')$ only potentially increases the elements of level $\ell'$, and the only change for the rest is a reset to the empty set. Therefore we only focus on showing the invariant for on $A_{i,\ell'}$. When LEVEL-CONSTRUCT$(\ell')$ is invoked, it iterates over all $i = 0 \ldots R$, and for each of them invokes BUCKET-CONSTRUCT on Line 8. By the definition, BUCKET-CONSTRUCT$(i', \ell')$ increments $S_{i',\ell'}$ and reduces $A_{i',\ell'}$ until the size of $A_{i',\ell'}$ becomes less than $2^{T-\ell'}$ or until $|S_{\mathrm{pred}(R,T)}| \geq k$ (see Line 6 of BUCKET-CONSTRUCT). After this invocation of BUCKET-CONSTRUCT the set $A_{i',\ell'}$ is not changed anymore. There are now two cases, depending on which of the two conditions on Line 6 of BUCKET-CONSTRUCT is false.

**Case $|S_{\mathrm{pred}(R,T)}| \geq k$.** In this case, each $A_{i,\ell'}$ is set to be the empty set (Line 11 of LEVEL-CONSTRUCT), and hence the claim follows directly.

**Case $|A_{i,\ell'}| < 2^{T-\ell'}$.** In this case, the size of $A_{i,\ell'}$ remains at most $2^{T-\ell'}$ throughout the rest of the execution. Since this holds for each $j = 0 \ldots R$ for which $|A_{j,\ell'}| < 2^{T-\ell'}$, after the loop on Line 2 terminates we have that $|A_{\ell'}| \leq (R+1)2^{T-\ell'}$. $\square$

**Lemma A.2** *If Invariants 1 and 2 hold before invoking* INSERTION *(Algorithm 3), then they also hold after* INSERTION *terminates.*

*Proof.* Observe that Line 1 of INSERTION changes only sets $B_\ell$. Hence, if Line 3 evaluates to false, then the two invariants still hold. Before we analyze the case when Line 3 evaluates to true, we first show that there does not exist $\ell'$ such that $|B_{\ell'}| > 2^{T-\ell'}$. Observe that INSERTION is the only function that adds elements to $B_{\ell'}$.

Towards a contradiction, assume that there is an invocation of INSERTION where for some $\ell'$ it holds that $|B_{\ell'}| > 2^{T-\ell'} \geq 1$. Let that be the $c$-th invocation of INSERTION. Since in each invocation of INSERTION the size of $B_{\ell'}$ increases by at most 1, it means that in the $(c-1)$-st invocation of INSERTION it holds that $|B_{\ell'}| \geq 2^{T-\ell'}$. Hence, Line 3 evaluates to true in that invocation. So, by the choice of $\ell^\star$ (see Line 4 of INSERTION) it holds that $\ell^\star \leq \ell'$. But this now implies that Line 6 of INSERTION sets $B_{\ell'}$ to be the empty set. Hence, after Line 3 of INSERTION evaluates to true in the $(c-1)$-st invocation, the size of $B_{\ell'}$ in the $c$-th invocation is at most 1. This contradicts our assumption.

This now implies that when LEVEL-CONSTRUCT is invoked, the two invariants hold. Hence, by Lemma A.1 these two invariants also hold after the execution of LEVEL-CONSTRUCT invoked on Line 7 of INSERTION, and consequently hold after the execution of INSERTION. $\square$

**Lemma A.3** *If Invariants 1 and 2 hold before invoking* DELETION *(Algorithm 4), then they also hold after* DELETION *terminates.*

*Proof.* On Lines 1 and 2 DELETION removes some elements from $A_{i,\ell}$ and $B_\ell$. So, these steps maintain Invariants 1 and 2. The rest of the changes of the sets $A_{i,\ell}$ and $B_\ell$ is done through invocation of LEVEL-CONSTRUCT on Line 9. The proof now follows by Lemma A.1. □

## A.1 Oracle-query Complexity

**Lemma A.4 (LEVEL-CONSTRUCT Complexity)** LEVEL-CONSTRUCT$(\ell)$ *performs* $O(R^4 \cdot 2^{T-\ell}/\epsilon_p)$ *oracle queries in expectation.*

*Proof.* Note that all the oracle queries performed by LEVEL-CONSTRUCT are via invocations of BUCKET-CONSTRUCT. Hence, we first analyze the oracle-query complexity of BUCKET-CONSTRUCT. We begin by bounding $|A_{i,\ell}|$ during an execution of LEVEL-CONSTRUCT$(\ell)$.

By the invariants, from Lines 4 and 6 of LEVEL-CONSTRUCT we have that for each $i$ and if $R \geq 2$ it holds that

$$
\begin{aligned}
|A_{i,\ell}| &\leq |B_{\ell-1}| + \sum_{j=0}^{R} |A_{j,\ell-1}| \\
&\leq 2^{T-\ell+1} + (R+1) \cdot 2^{T-\ell+1} \\
&\leq 4 \cdot R \cdot 2^{T-\ell}.
\end{aligned}
\tag{1}
$$

**Oracle queries of BUCKET-CONSTRUCT$(i,\ell)$.** For each $i$, LEVEL-CONSTRUCT invokes BUCKET-CONSTRUCT on Line 8. Line 2 of BUCKET-CONSTRUCT performs at most $|A_{i,\ell}|$ oracle queries. By Lemma C.4, PEELING invoked on Line 4 requires at most $c \cdot \log^2 k$ oracle queries, for some absolute constant $c$. Furthermore, from Item 4 of Lemma C.4 and our bound (1), BUCKET-CONSTRUCT in expectation executes at most $c_1 \cdot \frac{\log R}{\epsilon_p}$ iterations, for some absolute constant $c_1$, until $|A_{i,\ell}| < 2^{T-\ell}$. This altogether implies that in expectation BUCKET-CONSTRUCT$(i,\ell)$ performs at most

$$
c_1 \cdot \frac{\log R}{\epsilon_p} R \cdot (|A_{i,\ell}| + c \cdot \log^2 k)
\tag{2}
$$

oracle queries.

**Total number of oracle queries.** Let $C$ be the expected number of oracle queries performed by LEVEL-CONSTRUCT$(\ell)$. Given that LEVEL-CONSTRUCT$(\ell)$ might recursively invoke LEVEL-CONSTRUCT$(\ell')$ for all $\ell < \ell' \leq T$, from our bounds above we have

$$
\begin{aligned}
C &\overset{\text{by (2)}}{\leq} \sum_{\ell'=\ell}^{T} \sum_{i=0}^{R} c_1 \cdot \frac{\log R}{\epsilon_p} \cdot (|A_{i,\ell}| + c \cdot \log^2 k) \\
&\overset{\text{by (1)}}{\leq} c_1(R+1)\frac{\log R}{\epsilon_p} \cdot \sum_{\ell'=\ell}^{T} (4 \cdot R \cdot 2^{T-\ell} + c \cdot \log^2 k).
\end{aligned}
$$

Using that $\sum_{\ell'=\ell}^{T} 2^{T-\ell} \leq 2 \cdot 2^{T-\ell}$, that $R \geq \log k$, and also that $(T-\ell+1) \cdot c \cdot \log^2 k \leq 2^{T-\ell} \cdot c \cdot \log^2 k$, from the last chain of inequalities we further derive

$$
C = O(R^4 \cdot 2^{T-\ell}/\epsilon_p),
$$

as desired. □

**Lemma A.5 (Amortized complexity per deletion)** *The amortized number of oracle queries per deletion is* $O\left(\frac{R^5 \log^2(n)}{\epsilon_p^2 \cdot \varepsilon}\right)$ *in expectation.*

We first give an intuition of why this lemma holds, and then provide a formal proof. Let us concentrate on a single bucket $(i,\ell)$. By construction, any element added to $S_{i,\ell}$ is added from a set $A_{i,\ell}$ of

size at least $2^{T-\ell}$. This happens in every iteration inside BUCKET-CONSTRUCT, of which there are $O\left(\log(n)/\epsilon_p\right)$ many with high probability because of Property 4 of Lemma C.4. Therefore, any given element to be removed is not very likely to have been in the solution set $S_{i,\ell}$. Since a recomputation is triggered once an $\varepsilon$ fraction of that set is removed (see Line 7), it is required to remove a large $(\Omega(\epsilon_p\varepsilon/\log(n)))$ fraction of elements of $A_{i,\ell}$ (which has size at least $2^{T-\ell}$) in expectation before this happens. Once recomputation is triggered, it costs $O(2^{T-\ell} \cdot R^4/\epsilon_p)$ oracle queries by Lemma A.4. So, the overall amortized oracle-query complexity per deletion is $O\left(\frac{R^4 \log(n)}{\epsilon_p^2 \cdot \varepsilon}\right)$. We conclude by summing up these contribution over all buckets $(i, \ell)$.

*Proof.* We now formalize these arguments. Fix $(i, \ell)$. We will analyze the way $S_{i,\ell}$ is constructed by BUCKET-CONSTRUCT$(i, \ell)$ from $A_{i,\ell}$. This is done iteratively within the loop on Line 1. Let $I$ be the number of iterations performed by BUCKET-CONSTRUCT. Denote by $A_{i,\ell}^t$ the set $A_{i,\ell}$ and by $S_{i,\ell}^t$ the set $S_{i,\ell}$ at the end of the $t$-th iteration, for $t = 1, 2, ..., I$. Let $S_{i,\ell}^0 \stackrel{\text{def}}{=} \emptyset$. Assume that $|A_{i,\ell}^1| \geq 2^{T-\ell}$, as otherwise $S_{i,\ell}$ is empty and hence no deletion affects $S_{i,\ell}$.

First we argue that $I \leq 32\log(n)/\epsilon_p$ with high probability. Let $F_t$ be the fraction of elements not filtered away in the $t$-th iteration (if $t > I$, set $F_t = 0$). Property 4 of Lemma C.4 implies that $F_t$ is at most $1 - \epsilon_p/8$ in expectation (regardless of the state before the $t$-th iteration). Thus we have

$$\mathbb{E}\left[F_1 \cdot F_2 \cdot ... \cdot F_{32\log(n)/\epsilon_p}\right] \leq (1 - \epsilon_p/8)^{4\log(n)\cdot 8/\epsilon_p} \approx (1 - \epsilon_p/8)^{4\log_{1-\epsilon_p/8}(n)} = n^{-4}$$

and by Markov's inequality, the probability that $I > 32\log(n)/\epsilon_p$, for which it is necessary that $F_1 \cdot F_2 \cdot ... \cdot F_{32\log(n)/\epsilon_p} \geq 1/|A_{i,\ell}^1|$, is at most $|A_{i,\ell}^1| \cdot n^{-4} \leq n^{-3}$.

In the rest of the proof, we will show that in expectation it is needed to remove $\frac{\epsilon_p \varepsilon 2^{T-\ell}}{132 \log(n)}$ elements from $A_{i,\ell}^1$ in order for Line 7 of DELETION to become true.

Define $D^t \stackrel{\text{def}}{=} S_{i,\ell}^t \setminus S_{i,\ell}^{t-1}$, for each $1 \leq i \leq I$. The set $D^t$ is obtained on Line 4 of BUCKET-CONSTRUCT by invoking PEELING on $A_{i,\ell}^t$. By Lemma C.4, this set is a subset of $A_{i,\ell}^t$ of cardinality $|D^t|$ chosen uniformly at random. Observe that $A_{i,\ell}^t$ depends on the choice of $S_{i,\ell}^{t-1}$; in particular, $A_{i,\ell}^t \cap S_{i,\ell}^{t-1} = \emptyset$ as long as $\tau_i > 0$. Nevertheless, the randomness used by PEELING to obtain $D^t$ from $A_{i,\ell}^t$ does not depend on the choice of $S_{i,\ell}^{t-1}$. For $r = 1, 2, ...,$ let $X_r$ be the chronologically first $r$ elements removed from $A_{i,\ell}^1$. Then

$$\mathbb{E}\left[\frac{|D^t \cap X_r|}{|D^t|} \ \bigg| \ A_{i,\ell}^t\right] = \frac{|A_{i,\ell}^t \cap X_r|}{|A_{i,\ell}^t|} \leq \frac{r}{2^{T-\ell}}$$

and thus

$$\mathbb{E}\left[\frac{|D^t \cap X_r|}{|D^t|}\right] \leq \frac{r}{2^{T-\ell}}.$$

We write

$$\mathbb{E}\left[\frac{|X_r \cap S_{i,\ell}|}{|S_{i,\ell}|}\right] = \mathbb{E}\left[\frac{|X_r \cap S_{i,\ell}|}{|S_{i,\ell}|} \ \bigg| \ I \leq \frac{32\log(n)}{\epsilon_p}\right] \cdot \mathbb{P}\left[I \leq \frac{32\log(n)}{\epsilon_p}\right]$$
$$+ \underbrace{\mathbb{E}\left[\frac{|X_r \cap S_{i,\ell}|}{|S_{i,\ell}|} \ \bigg| \ I > \frac{32\log(n)}{\epsilon_p}\right]}_{\leq 1} \cdot \underbrace{\mathbb{P}\left[I > \frac{32\log(n)}{\epsilon_p}\right]}_{\leq n^{-3}}.$$

Let us adopt the convention that $D^t = \emptyset$ and $\frac{|D^t \cap X_r|}{|D^t|} = 0$ for $t > I$. Then

$$\mathbb{E}\left[\frac{|X_r \cap S_{i,\ell}|}{|S_{i,\ell}|} \,\bigg|\, I \le \frac{32 \log(n)}{\epsilon_p}\right] = \mathbb{E}\left[\sum_{t=1}^{I} \frac{|X_r \cap D^t|}{|S_{i,\ell}|} \,\bigg|\, I \le \frac{32 \log(n)}{\epsilon_p}\right]$$

$$\le \sum_{t=1}^{32 \log(n)/\epsilon_p} \mathbb{E}\left[\frac{|X_r \cap D^t|}{|D^t|} \,\bigg|\, I \le \frac{32 \log(n)}{\epsilon_p}\right]$$

$$\le \sum_{t=1}^{32 \log(n)/\epsilon_p} \mathbb{E}\left[\frac{|X_r \cap D^t|}{|D^t|}\right] \cdot \mathbb{P}\left[I \le \frac{32 \log(n)}{\epsilon_p}\right]^{-1}$$

$$\le \frac{32 \log(n)}{\epsilon_p} \cdot \frac{r}{2^{T-\ell}} \cdot \mathbb{P}\left[I \le \frac{32 \log(n)}{\epsilon_p}\right]^{-1}$$

(for the second inequality we used the simple fact that $\mathbb{E}[X \mid A] \le \mathbb{E}[X]/\mathbb{P}[A]$ for $X \ge 0$). In the end we get

$$\mathbb{E}\left[\frac{|X_r \cap S_{i,\ell}|}{|S_{i,\ell}|}\right] \le \frac{32 \log(n)}{\epsilon_p} \cdot \frac{r}{2^{T-\ell}} + n^{-3}.$$

Let $D$ denote the number of elements removed before Line 7 of DELETION evaluates to true. We have

$$\mathbb{E}[D] = \sum_{r=0}^{\infty} \mathbb{P}[D > r]$$

$$\ge \sum_{r=1}^{\frac{\varepsilon 2^{T-\ell}}{66 \log(n)/\epsilon_p}} \mathbb{P}\left[\frac{|X_r \cap S_{i,\ell}|}{|S_{i,\ell}|} < \varepsilon\right]$$

$$\ge \sum_{r=1}^{\frac{\varepsilon 2^{T-\ell}}{66 \log(n)/\epsilon_p}} \underbrace{\left(1 - \frac{32 \log(n)/\epsilon_p \cdot \frac{r}{2^{T-\ell}} + n^{-3}}{\varepsilon}\right)}_{\ge 1 - \frac{33 \log(n)/\epsilon_p \cdot \frac{r}{2^{T-\ell}}}{\varepsilon} \ge 1/2}$$

$$\ge \frac{\epsilon_p \varepsilon 2^{T-\ell}}{132 \log(n)},$$

where the second-last inequality follows from Markov's inequality. Finally, a recomputation costs $O(2^{T-\ell} \cdot R^4/\epsilon_p)$ oracle queries by Lemma A.4. This is in expectation over the randomness used in the recomputation, which is independent from the randomness used to determine $D$ (so we can compute the expectation of the ratio using the ratio of the expectations). Thus the expected amortized cost per deleted element is $O\left(\frac{R^4 \log(n)}{\epsilon_p^2 \varepsilon}\right)$.

We obtain the final bound by summing up the contributions of all $RT$ buckets to this amortized expected recomputation cost. $\square$

**Lemma A.6 (Amortized complexity per insertion)** *The amortized number of oracle queries per insertion is $O(T \cdot R^4/\epsilon_p)$ in expectation.*

*Proof.* When an element $e$ is inserted, it is added to $B_\ell$ for all $0 \le \ell \le T$ (see Line 1 of INSERTION).

Assume that after adding $e$ some sets $B_\ell$ become "too large", i.e., $|B_\ell| \ge 2^{T-\ell}$ (see Line 3). Let $\ell^\star$ be the smallest such $\ell$. Then, INSERTION invokes LEVEL-CONSTRUCT($\ell^\star$) on Line 7. By Lemma A.4, this invocation requires $O(2^{T-\ell^\star} \cdot R^4/\epsilon_p)$ oracle queries in expectation. Also, during this invocation the set $B_{\ell^\star}$ is set to be the empty set (see Line 1 of LEVEL-CONSTRUCT($\ell^\star$)). Moreover, at the beginning of the algorithm $B_{\ell^\star}$ was empty and is augmented only by INSERTION. This altogether means that $2^{T-\ell^\star}$ elements have to be added to $B_{\ell^\star}$ in order for INSERTION to invoke this execution of LEVEL-CONSTRUCT. This implies that per one element added to $B_{\ell^\star}$, INSERTION uses $O(R^4/\epsilon_p)$

oracle queries. Moreover, an element is added to $T$ different sets $B_\ell$. Therefore, across all $\ell$, INSERTION in expectation spends $O(T \cdot R^4/\epsilon_p)$ oracle queries per one inserted element. $\square$

Together, Lemmas A.5 and A.6 imply the following result.

**Theorem 5.2** *The amortized expected number of oracle queries per update is* $O\left(\frac{R^5 \log^2(n)}{\epsilon_p^2 \cdot \varepsilon}\right)$, *where* $R$ *equals* $\log_{1+\epsilon_1}(2k)$ *(see Algorithm 1).*

# B   Correctness of our algorithm

Let us start by introducing a key property of our algorithm. Throughout this section, in case $\ell = 0$, by $S_{\mathrm{pred}(r,\ell-1)}$ we denote $S_{\mathrm{pred}(r-1,T)}$ and we define $S_{\mathrm{pred}(-1,T)} = \emptyset$.

**Observation B.1** *The only time when elements are added to one of the sets $S_{i,\ell}$ is Line 4 of Algorithm 2. Moreover, all the sets $S_{\cdot,\cdot}$ after $S_{i,\ell}$ are empty, i.e.,*

$$\bigcup_{0 \leq j \leq R, \ell < r \leq T} S_{j,r} \cup \bigcup_{i+1 \leq j \leq R} S_{j,\ell} = \emptyset.$$

*Proof.* This follows from the fact that we empty all the abovementioned sets in Line 5 of Algorithm 3 and Line 8 of Algorithm 4 before calling LEVEL-CONSTRUCT, and those are the only lines that BUCKET-CONSTRUCT is called from. $\square$

**Theorem 5.1** *Let $Sol_i$ be the solution of our algorithm and $\mathrm{OPT}_i$ be the optimal solution after $i$ updates. Moreover, assume that $\gamma$ in Algorithm 1 is such that $(1 + \epsilon_p) \mathrm{OPT}_i \geq \gamma \geq \mathrm{OPT}_i$. Then for any $1 \leq i \leq n$ we have $\mathbb{E}[f(Sol_i)] \geq (1 - \epsilon_p - \epsilon(1+\epsilon_1)) \frac{\mathrm{OPT}_i}{2}$.*

*Proof.* Consider the last time $j$ when LEVEL-CONSTRUCT$(\ell)$ is called for some value $\ell$. Let us start by analysing $f(Sol_j)$. Consider the following two cases depending on the size of $S_{\mathrm{pred}(R,T)}$ at this moment, i.e., after the $j$-th operation:

- If $|S_{\mathrm{pred}(R,T)}| = k$: then by Item 3 of Lemma C.4 any set of elements that are added to one of the sets $S$ gives marginal contribution of $\tau_i$ per element to the previous elements of $S$

$$\mathbb{E}[f(S_{i,\ell}|S_{\mathrm{pred}(i,\ell-1)})] \geq (1 - \epsilon_p) \tau_i |S_{i,\ell}| \geq (1 - \epsilon_p) \frac{\gamma}{2k} |S_{i,\ell}|.$$

  Therefore since in this case $|Sol_j| = k$, and by Observation B.1, by linearity of expectation we get:

$$\mathbb{E}[f(Sol_j)] \geq k \cdot (1 - \epsilon_p) \frac{\gamma}{2k} \geq (1 - \epsilon_p) \frac{\mathrm{OPT}_i}{2}. \tag{3}$$

- If $|S_{\mathrm{pred}(R,T)}| < k$: The goal in this case is to show that for any element $e \in O_j$, $f(e|S_{\mathrm{pred}(R,T)}) < \frac{\gamma}{2k}$. To that end, consider any element $e \in O_j$. Consider the LEVEL-CONSTRUCT$(\ell')$ with the lowest $\ell'$ that is called after inserting this element[10]. In Line 4 or Line 6 of the LEVEL-CONSTRUCT algorithm, $e$ will be added to some sets in $A$. Moreover, in this case, LEVEL-CONSTRUCT$(T)$ must be called, since this is the only way how the condition in Line 13 can be false. This also results in calling BUCKET-CONSTRUCT$(r,T)$ for all $1 \leq r \leq R$ in Line 2 of Algorithm 5. Notice that BUCKET-CONSTRUCT$(r,T)$ stops only if $|A_{r,T}| = 0$ (in Line 6). Therefore all the elements have been removed from the sets $A$ at some point. Also, the only time when we remove elements from $A$ is in Line 2, which combined with submodularity shows that for any element $e \in O_j$ we have

$$f(e|S_{\mathrm{pred}(R,T)}) \leq \tau_R = \frac{\gamma}{2k}.$$

By definition, $\mathrm{Sol}_j = S_{\mathrm{pred}(R,T)}$, which results in

$$f(e|\mathrm{Sol}_j) \leq \frac{\gamma}{2k}.$$

Applying the above inequality for all the elements $e \in O_j$ along with submodularity, we get that

$$f(O_j|\mathrm{Sol}_j) \leq k \cdot \frac{\gamma}{2k} \leq \frac{\gamma}{2}.$$

Moreover, by submodularity and monotonicity, we have that

$$f(O_j) \leq f(\mathrm{Sol}_j) + f(O_j|\mathrm{Sol}_j).$$

Combining the above two inequalities we get that

$$f(\mathrm{Sol}_j) \geq \mathrm{OPT}_j - \frac{\gamma}{2} \geq \mathrm{OPT}_j - \frac{1+\epsilon_p}{2}\mathrm{OPT}_i \geq \frac{1-\epsilon_p}{2}\mathrm{OPT}_i, \qquad (4)$$

where the last two inequalities follow by the theorem's assumption and the fact that $\mathrm{OPT}_j \geq \mathrm{OPT}_i$, respectively. Notice that $\mathrm{OPT}_j \geq \mathrm{OPT}_i$ since there are no insertions after the last call to LEVEL-CONSTRUCT.

By (3) and (4) we get that

$$\mathbb{E}[f(\mathrm{Sol}_j)] \geq (1-\epsilon_p)\frac{\mathrm{OPT}_i}{2}. \qquad (5)$$

Let us now complete the proof by showing that $f(\mathrm{Sol}_i) \geq (1 - \frac{\epsilon(1+\epsilon_1)}{1-\epsilon_p})f(\mathrm{Sol}_j)$ for $i > j$. Notice that by the assumption that $\gamma \geq \mathrm{OPT}$ there is no element $e$ with $f(e) > \gamma$. Therefore all the elements on any layer will belong to one of the buckets. Consider any $S_{r,\ell}$ ($1 \leq r \leq R, 0 \leq \ell \leq T$); we know that when LEVEL-CONSTRUCT($\ell$) has been called, we have $E[f(S_{r,\ell}|S_{\mathrm{pred}(r,\ell-1)})] \geq (1-\epsilon_p)\tau_r|S_{r,\ell}|$.[11] Moreover, at most an $\epsilon$-fraction of its elements can be removed. By $S'_{r,\ell}$ we denote this set after these deletions. We know that the marginal contribution of each element in $S_{r,\ell}$ with respect to $S_{\mathrm{pred}(r,\ell-1)}$ is at most $\tau_{i-1} = (1+\epsilon_1)\tau_i$. By submodularity we get that

$$f(S'_{r,\ell}|S_{\mathrm{pred}(r,\ell-1)}) \geq f(S_{r,\ell}|S_{\mathrm{pred}(r,\ell-1)}) - |S_{r,\ell}|\epsilon(1+\epsilon_1)\tau_i f(S'_{r,\ell}|S_{\mathrm{pred}(r,\ell-1)})$$

$$\geq f(S_{r,\ell}|S_{\mathrm{pred}(r,\ell-1)})(1 - \frac{\epsilon(1+\epsilon_1)}{1-\epsilon_p}).$$

Now for all $r$ and $\ell$, let $S'_{\mathrm{pred}(r,\ell)}$ denote the set $S_{\mathrm{pred}(r,\ell)}$ after the $i$-th operation, i.e., after applying the deletions.

Considering that $f$ is submodular, and $S'_{\mathrm{pred}(r,\ell)} \subseteq S_{\mathrm{pred}(r,\ell)}$, we get

$$f(S'_{r,\ell}|S'_{\mathrm{pred}(r,\ell-1)}) \geq f(S_{r,\ell}|S_{\mathrm{pred}(r,\ell-1)})(1 - \frac{\epsilon(1+\epsilon_1)}{1-\epsilon_p}).$$

By adding up the above marginal values over $r$ and $\ell$, we get

$$f(S'_{\mathrm{pred}(R,T)}) \geq (1 - \frac{\epsilon(1+\epsilon_1)}{1-\epsilon_p})f(S_{\mathrm{pred}(R,T)}).$$

So $f(\mathrm{Sol}_i) \geq (1 - (1 - \frac{\epsilon(1+\epsilon_1)}{1-\epsilon_p}))f(\mathrm{Sol}_j)$. This along with Eq. (5) concludes the proof:

$$f(\mathrm{Sol}_i) \geq (1 - \frac{\epsilon(1+\epsilon_1)}{1-\epsilon_p})f(\mathrm{Sol}_j),$$

$$\mathbb{E}[f(\mathrm{Sol}_i)] \geq (1 - \epsilon_p - \epsilon(1+\epsilon_1))\frac{\mathrm{OPT}_i}{2}.$$

$\square$

# C Peeling algorithm

For completeness, in this section we present the routine PEELING, which is part of the algorithm THRESHOLD-SAMPLING in [FMZ19]. All the lemmas and algorithms in this section have been introduced in [FMZ19] and we provide them here for completeness. Before presenting PEELING, we need to define a distribution. Let $N \subseteq V$ be some set of elements.

**Definition C.1** *Conditioned on the current state of the algorithm, consider the process where first a set $S \sim \mathcal{U}(N, s)$ (size-$s$ subset of $N$) and then an element $x \sim N \setminus S$ are drawn uniformly at random. Let $\mathcal{D}_s$ denote the probability distribution over the indicator random variable*

$$I_s = \mathbb{1}[f(x \mid S) \geq \tau].$$

---

**Algorithm 6** PEELING

---

**Input:** Subset of items $N \subseteq V$, function $f : 2^N \to R^{\geq 0}$, constraint $k$, threshold $\tau$, error $\epsilon$

1: Set smaller error $\hat{\epsilon} \leftarrow \epsilon/4$
2: Set $m \leftarrow \lceil \log(k)/\hat{\epsilon} \rceil$
3: **for** $i = 0$ to $m$ **do**
4:      Set $s \leftarrow \min\{\lfloor (1 + \hat{\epsilon})^i \rfloor, |N|\}$
5:      **if** REDUCED-MEAN$(\mathcal{D}_s, \hat{\epsilon})$ **then**
6:          **break**
7:      **end if**
8: **end for**
9: Sample $S \sim \mathcal{U}(N, \min\{s, k\})$
10: **return** $S$

---

We briefly remark that the REDUCED-MEAN subroutine is a standard unbiased estimator for the mean of a Bernoulli distribution. Since $\mathcal{D}_t$ is a uniform distribution over indicator random variables, it is in fact a Bernoulli distribution. The guarantees of in Lemma C.2 are consequences of Chernoff bounds [BS06].

---

**Algorithm 7** REDUCED-MEAN

---

**Input:** access to a Bernoulli distribution $\mathcal{D}$, error $\hat{\epsilon}$

1: Set failure probability $\delta \leftarrow 2\hat{\epsilon}^2/(k \log(k))$
2: Set number of samples $m \leftarrow 16\lceil \log(2/\delta)/\hat{\epsilon}^2 \rceil$
3: Sample $X_1, X_2, \ldots, X_m \sim \mathcal{D}$
4: Set $\overline{\mu} \leftarrow \frac{1}{m}\sum_{i=1}^m X_i$
5: **if** $\overline{\mu} \leq 1 - 1.5\hat{\epsilon}$ **then**
6:     **return** true
7: **end if**
8: **return** false

---

**Lemma C.2** *For any Bernoulli distribution $\mathcal{D}$, REDUCED-MEAN uses $O(\log(\delta^{-1})/\hat{\epsilon}^2)$ samples to correctly report one of the following properties with probability at least $1 - \delta$:*

    *1. If the output is* true*, then the mean of $\mathcal{D}$ is $\mu \leq 1 - \hat{\epsilon}$.*

    *2. If the output is* false*, then the mean of $\mathcal{D}$ is $\mu \geq 1 - 2\hat{\epsilon}$.*

*Here $\delta$ is set to be $2\hat{\epsilon}^2/(k \log(k))$ in the algorithm REDUCED-MEAN.*

*Proof.* By construction, the number of samples is $m = 16\lceil \log(2/\delta)/\hat{\epsilon}^2 \rceil$. To show the correctness of REDUCED-MEAN, it suffices to prove that $\Pr(|\overline{\mu} - \mu| \geq \hat{\epsilon}/2) \leq \delta$. Letting $X = \sum_{i=1}^m X_i$, this is equivalent to

$$\Pr\left(|X - m\mu| \geq \frac{\hat{\epsilon}m}{2}\right) \leq \delta.$$

Using the Chernoff bounds in Lemma C.3 and a union bound, for any $a > 0$ we have

$$\Pr(|X - m\mu| \geq a) \leq e^{-\frac{a^2}{2m\mu}} + e^{-a\min\left(\frac{1}{5}, \frac{a}{4m\mu}\right)}.$$

Let $a = \hat{\epsilon}m/2$ and consider the exponents of the two terms separately. Since $\mu \leq 1$, we bound the left term by

$$\frac{a^2}{2m\mu} = \frac{\hat{\epsilon}^2 m^2}{8m\mu} \geq \frac{\hat{\epsilon}^2}{8\mu} \cdot \frac{16\log(2/\delta)}{\hat{\epsilon}^2} \geq \log(2/\delta).$$

For the second term, first consider the case when $1/5 \leq a/(4m\mu)$. For any $\hat{\epsilon} \leq 1$, it follows that

$$a\min\left(\frac{1}{5}, \frac{a}{4m\mu}\right) = \frac{1}{5} \geq \frac{\hat{\epsilon}}{10} \cdot \frac{16\log(2/\delta)}{\hat{\epsilon}^2} \geq \log(2/\delta).$$

Otherwise, we have $a/(4m\mu) \leq 1/5$, and by the previous analysis we have $a^2/(4m\mu) \geq \log(2\delta)$. Therefore, in all cases we have

$$\Pr\left(|X - m\mu| \geq \frac{\hat{\epsilon}m}{2}\right) \leq 2e^{-\log(2/\delta)} = \delta,$$

which completes the proof. □

**Lemma C.3 (Chernoff bounds, [BS06])** *Suppose $X_1, \ldots, X_n$ are binary random variables such that $\Pr(X_i = 1) = p_i$. Let $\mu = \sum_{i=1}^{n} p_i$ and $X = \sum_{i=1}^{n} X_i$. Then for any $a > 0$, we have*

$$\Pr(X - \mu \geq a) \leq e^{-a\min\left(\frac{1}{5}, \frac{a}{4\mu}\right)}.$$

*Moreover, for any $a > 0$, we have*

$$\Pr(X - \mu \leq -a) \leq e^{-\frac{a^2}{2\mu}}.$$

Using the guarantees for REDUCED-MEAN, we can prove:

**Lemma C.4** *Let $N$ be a set of elements such that for each $e \in N$ we have $f(e) \geq \tau$. The algorithm PEELING outputs a set $S \subseteq N$ with $|S| \leq k$ such that the following properties hold:*

1. *There are $O(\log^2(k))$ oracle queries.*

2. *PEELING finds a size $X$ and returns a uniformly random subset of size $X$ from its input items.*

3. *The expected average marginal satisfies $\mathbb{E}[f(S)/|S|] \geq (1 - \epsilon_p)\tau$.*

4. *If $|S| < k$, then the expected number of items $x \in N$ with $\Delta(x, S) < \tau$ is at least $\epsilon_p|N|/8$.*

*Proof.*

We prove the upper bound on the query complexity of PEELING. There are $m = O(\log(k))$ runs of REDUCED-MEAN, each of which makes $O(\delta^{-1}) = O(\log(k))$ queries. Therefore the total query complexity of PEELING is $O(\log^2(k))$.

Next, we show the lower bound on the average value of selected items, the set $S$. PEELING starts by calling REDUCED-MEAN with $s = 1$. We note that in our case the input of PEELING always consists of items with marginal value at least $\tau$. Therefore the first run of REDUCED-MEAN returns false. We call REDUCED-MEAN with different values of $s$. Let $s'$ be the first time REDUCED-MEAN returns true and $s''$ be the previous value that we called REDUCED-MEAN with. If REDUCED-MEAN always returns false, we let $s''$ be the maximum value of $s$.

We call REDUCED-MEAN $\lceil \log(k)/\hat{\epsilon} \rceil \leq 2\log(k)/\hat{\epsilon}$ times. Using Lemma C.2 and a union bound, we know that with probability at least $1 - 2\delta\log(k)/\hat{\epsilon} \geq 1 - \hat{\epsilon}/k$ for all calls of REDUCED-MEAN we have the two properties of Lemma C.2.

For $s''$, REDUCED-MEAN returns false, therefore the mean of random variable $\mathcal{D}_{s''}$ is at least $1 - 2\hat{\epsilon}$. So picking $s''$ random items yields at least $(1 - 2\hat{\epsilon})\tau s''$ value. Here we use linearity of expectation, and also use that the expected marginal gain of a randomly chosen element with respect to a random subset $Z \subseteq V$ does not increase with the increase of size of $Z$. (We refer a reader to Lemma 3.4 and Lemma 3.2 of [FKK18] for a formal proof of this argument.) PEELING returns a random subset of size $X = \min(s, k)$. By definition of $s''$, we have $X \leq (1 + \hat{\epsilon})s''$. Therefore the expected value of the solution set $S$ is at least $\frac{1-2\hat{\epsilon}}{1+\hat{\epsilon}}|S|\tau$. The above statements hold only if we are in the case where REDUCED-MEAN does not fail in any of the calls. Since the failure probability is upper-bounded by $\hat{\epsilon}/k$ and even in the failure case we do not pick more than $k$ items, we can still say that the expected value of the solution is at least $(1 - \hat{\epsilon})\frac{1-2\hat{\epsilon}}{1+\hat{\epsilon}}|S|\tau \geq (1 - \epsilon)\tau|S|$. The inequality holds because of the way the parameter $\hat{\epsilon}$ is set. This proves the lower bound on the expected value of the solution.

To prove the last property of the lemma, we note that if the solution set $S$ consists of fewer than $k$ items, we know that REDUCED-MEAN has returned true at least once. Denote by $s'$ the first time it does so. In this case, PEELING returns a random set of size $s'$. We know that REDUCED-MEAN did not fail in any of the calls with probability at least $1 - \hat{\epsilon}/k$. Focusing on the case that REDUCED-MEAN does not fail, we know that the mean of $\mathcal{D}_{s'}$ is at most $1 - \hat{\epsilon}$. Therefore after picking solution $S$ (e.g., $s'$ random items), the expected number of items with marginal value below $\tau$ is at least an $\hat{\epsilon}$-fraction of all input items. Noting that the failure probability is at most $1 - \hat{\epsilon}/k \leq 1/2$ proves the last claim of the lemma.

$\square$

## C.1   Combining the ingredients

We now recall well-known techniques that can be used to remove the assumptions we made while designing our algorithm. First, we assumed that we have a tight estimate of OPT, e.g., the value of $\gamma$ in Theorem 5.1. This assumption can be removed by considering geometrically increasing guesses $\gamma = (1 + \epsilon_p)^i$ of OPT, and for each of the guesses executing a separate instance of our algorithm. Even though OPT potentially changes from operation to operation, at each point one of the guesses is correct up to a small multiplicative factor. A similar approach was employed in several prior works. After every operation, we return the maximum-value solution over all $\gamma$'s. Theorem 5.1 shows that, for the $\gamma$ value that is close to the true optimum value at that time, the instance parametrized by $\gamma$ returns a solution of high value. Moreover, the number of oracle calls is independent of the value of $\gamma$. This results in losing a factor $\log(k\Delta/(\delta\epsilon_p))$ in the number of oracle calls, where $\Delta, \delta$ denote the value of the elements of maximum and minimum value in the universe, respectively. We do not need to know these two values in advance; we simply compute them on the fly and run parallel copies of the algorithm for the currently relevant guesses of OPT. Moreover, we can again use a simple technique to remove the dependency on $\log(\Delta/\delta)$. Namely, it suffices to add an element $e$ to those copies of the algorithm where $f(e) \leq \gamma \leq \frac{k}{\epsilon_p}f(e)$ since: (i) while $e$ is not deleted it holds OPT $\geq f(e)$, therefore we do not need to consider copies with $\gamma < f(e)$; and, (ii) all elements with $\gamma \geq \frac{k}{\epsilon_p}f(e)$ contribute at most only $\epsilon_p\gamma$ to the solution of this copy. Therefore, this increases the number of oracle calls by a factor of $\log k/\epsilon_p$, while decreasing the approximation guarantee by $1 - \epsilon_p$.

Second, we assumed that we know the length $n$ of the stream, which is used to upper-bound $|V_t|$. We remove this assumption as follows. We maintain an upper-bound $\tilde{n}$ on $n$. The algorithm is initiated by $\tilde{n} = 1$. If at some point $n$ equals $\tilde{n}$, we restart the algorithm by doubling $\tilde{n}$, i.e, by letting $\tilde{n} \leftarrow 2 \cdot \tilde{n}$, and defining $T = \log \tilde{n}$ in Algorithm 1. This affects the number of oracle calls only by a constant factor, and has no effect on the approximation guarantee.

**Theorem 5.3** *Our algorithm maintains a $(1 - 2\epsilon_p - \epsilon(1 + \epsilon_1))/2$-approximate solution after each operation. The amortized expected number of oracle queries per update of this algorithm is $O\left(\frac{\log_{1+\epsilon_1}^6(k)\log^2(n)}{\epsilon_p^4 \cdot \varepsilon}\right)$.*

Figure 2: These plots depict results of our experiments run on Twitter network, with the elements presented as an arbitrary ordered stream. The optimization is performed over windows of size $70,000$.

## D  Additional experiments

In this section, we provide additional experiments to those presented in Fig. 1. The setup we use in this section is the same as in Section 6. We refer a reader to Section 6 for the details of this setup (e.g., definition of $f$ and the type of plots we present).

We perform two types of experiments:

- plotting the number of oracle calls and the values of $f$ with respect to $k$, while varying $k$ (as the plots in Fig. 1), and

- plotting the number of oracle calls and the values of $f$ for blocks of queries (insertions and deletions) for a *fixed $k$* (as the plots in Fig. 9).

**Plots for varying $k$.**  We perform the following experiments:

- **ego-Twitter:** In Fig. 2 we depict the result of our experiments performed on the ego-Twitter network where the nodes are processed as arbitrary ordered stream. We perform optimization over windows of size $70,000$. Recall that for this dataset $|V| = 81,306$.

- **Pokec:** The plots in Figs. 3 and 4 are obtained on the dataset Pokec. In this case, we run two experiments: experiments over window of size $1,200,000$ (see Fig. 3); and an experiment where all the nodes are inserted first and then those with largest neighborhoods are deleted (see Fig. 4). Recall that for this dataset $|V| = 1,632,803$.

**Results.**  In each of the experiments the quality of outputs of our algorithm matches those of SIEVESTREAMING. In term of the number of oracle calls, our algorithm performs at most as many as SIEVESTREAMING, while most often our method performs significantly fewer. It is interesting to note that in Fig. 4 $\text{ALG}_{0.2}$ performs significantly fewer oracle calls than $\text{ALG}_{0.0}$, while in the same time not losing on the quality of output.

In the sliding-window setting, where CNZ is applicable, we observe that the quality of the output of our approach it similar the one of CNZ. In Fig. 3 the baseline $\text{CNZ}_{0.2}$ performs almost $50\%$ fewer oracle calls than our approach, at the expenses of outputting solutions of around $5\%$ worse quality compared to $\text{ALG}_{0.2}$ and $\text{ALG}_{0.0}$.

**Plots for fixed $k$.**  In this type of experiments we fix the value of $k$ and split all the operations (insertions/deletions) performed in a given experiment into $400$ blocks, all blocks representing the same number of operations. Then for each block we either plot the number of oracle calls or plot the average value of $f$. All the plots are obtained for Enron dataset.

**Results.**  In Fig. 5, we give results Enron for $k = 20$ for window of size $30,000$, which is a more detailed experiment of the one in Fig. 1 (a) for $k = 20$. We plot the number of oracle calls performed in each block (Fig. 5 (a)), and also the cumulative number of oracle calls performed up to each block

Figure 3: These plots depict results of our experiments run on Pokec network, with the elements presented as an arbitrary ordered stream. The optimization is performed over windows of size $1,200,000$.

Figure 4: These plots represent the execution of our algorithm on the Pokec network. The nodes of this network are first inserted in arbitrary order, and then all the nodes are deleted by deleting first those having largest neighborhood.

(Fig. 5 (b)). The same type of experiment is performed for $f$ as well and depicted in Fig. 9. Similar results for $k = 50$ and $k = 70$ are given in Fig. 6.

For the results on number of oracle calls, we observe that there exists a small number of operations over which our algorithm performs a significant number of oracle calls. This suggest that indeed, as we state in our main theorem, our algorithm has only amortized poly-logarithmic update time. This also leads to an interesting open question to design an algorithm that requires poly-logarithmic worst case update time. Despite this, our algorithm in total performs fewer oracle calls than $CNZ_{0.1}$, $CNZ_{0.2}$ and SIEVESTREAMING, while significantly outperforming SIEVESTREAMING.

In terms of $f$, the plots in Fig. 6 show the same behavior as the plot in Fig. 9.

In Fig. 7 we show results of experiments on Enron where the nodes of this network are first inserted in arbitrary order, and then all the nodes are deleted by deleting first those having largest neighborhood. It is interesting to note that for these experiments there are no operations where ALG has significant increase in the number of calls as it has for window-experiments.

### D.1 Value of $f$ after each operation.

Fig. 1 shows the average value of $f$ for different values of $k$ over all operations (insertions/deletions). In Fig. 9 we present a more detailed view of the experiment of Fig. 1 (b) in the following sense. We fix $k = 20$ and split the operations into $400$ equal-sized blocks. For each block, we compute the average value of $f$. We plot those values for all the baselines. This experiment allows us to compare our approach and the baselines over the course of the entire execution. We can see that ALG is very similar to $CNZ_{0.1}$ in each block, while $CNZ_{0.2}$ shows around $10\%$ worse performance in the blocks where $f$ has the highest value. In those blocks, SIEVESTREAMING has around $5\%$ better performance than ALG and $CNZ_{0.1}$.

Figure 5: This plot present more detailed analysis of Fig. 1 (a) for $k = 20$ which, as a reminder, depicts a window-experiment results performed on Enron. The whole execution (i.e., the performed operations) used to obtain the point in Fig. 1 (a) for $k = 20$ is divided into 400 blocks. Plot (a) shows count of oracle calls in each block separately, while plot (b) shows cumulative number of oracle calls. Legend is the same for both plots in this figure.

Figure 6: The setup in this figure is the same as the one for Figs. 5 and 9, and extends the result from Fig. 1 (a) and (b) for $k = 50$ (plots (a) and (b)) and $k = 70$ (plots (c) and (d)). Legend is the same for all plots in this figure.

Figure 7: These plots represent the execution of our algorithm on Enron. The nodes of this network are first inserted in arbitrary order, and then all the nodes are deleted by deleting first those having largest neighborhood. Plots (a) and (b) correspond to $k = 20$, plots (c) and (d) correspond to $k = 50$, and plots (e) and (f) correspond to $k = 70$. The whole execution (i.e., the performed operations) for each experiment is divided into $400$ blocks. Plots (a), (c) and (e) show average $f$ in each block separately, while plots (b), (d) and (f) show cumulative number of oracle calls. Legend is the same for all plots in this figure.

# E   Details of the implemented algorithm

In this section we explain a simpler version of our algorithm, which we use for the implementation and the experiments. We do this as we believe that the bucketing idea is not crucial in real-world applications, even though it is needed to achieve the theoretical guarantee. Namely, given the random structure of layers described in our algorithm, it is very hard for an adversary to delete good elements in a layer without deleting many elements, as the notion of "good" (i.e., contribution of the element with respect to previously chosen elements) heavily depends on the random elements chosen in the previous layers. We believe this situation does not happen often in practice, and so we can simply treat all the elements in a layer in the same way. The only difference is that in this algorithm we do not partition the elements with respect to their contribution and only consider one bucket in each layer. In what follows, we present the algorithm in details for the sake of completeness. One can also read this section without reading the main algorithm.

Similar to before, in this section we assume we know OPT and first present an algorithm for the case when deletions appear only after all the insertions have been performed. Later, we explain a slight

Figure 8: Plots (a), (b), (c) and (d) in this figure depict standard deviation in percentage of the values ploted in Fig. 1 (a), (b), (c) and (d), respectively.

Figure 9: This plot presents a more detailed analysis of Fig. 1 (b) for $k = 20$; recall that this corresponds to a window-experiment results performed on Enron. The entire execution (i.e., the performed operations) used to obtain the point in Fig. 1 (b) for $k = 20$ is divided into 400 blocks. This plot shows the average value of $f$ within each block.

modification to our algorithm to support fully dynamic insertions and deletions. In the following we refer to the set of inserted elements as $V$.

We will construct a hierarchy of sets $H_1, H_2, \ldots$. Let $H_1$ be all the elements $e \in V$ such that $f(e) \geq \frac{\text{OPT}}{2k}$, where OPT is the value of the optimum solution of the set $V$. We first define

$$H_1 \leftarrow \left\{ e \in V \ \middle| \ f(e) \geq \frac{\text{OPT}}{2k} \right\}.$$

Afterwards, we select an element $e_1$ from $H_1$ uniformly at random and we define $H_2$ as

$$H_2 \leftarrow \left\{ e \in H_1 \ \middle| \ f(e|\{e_1\}) \geq \frac{\text{OPT}}{2k} \right\}.$$

We repeat this procedure until either we have chosen $k$ elements or for some $1 \leq \ell \leq k$, the set $H_\ell$ has become empty. More precisely, for any $2 \leq \ell \leq k$, we define

$$H_\ell \leftarrow \left\{ e \in H_{\ell-1} \ \middle| \ f(e|\{e_1, \ldots, e_{\ell-1}\}) \geq \frac{\text{OPT}}{2k} \right\}.$$

Then we let $e_\ell$ be a random element from $H_\ell$. We call this simple procedure *a round of peeling*. It is easy to see that the running time of this simple procedure is $O(k|V|)$. This means that the average running time is $O(k)$ per element. Let $m$ be the number of elements we have picked this way. Now we can observe that the solution $E = \{e_1, \ldots, e_m\}$ is a $1/2$-approximation:

- If $|E| = k$, then simply by adding the marginal contributions we get that:

$$f(E) = \sum_{1 \leq \ell \leq k} f(e_\ell | e_1 \ldots e_{\ell-1}) \geq k \frac{\text{OPT}}{2k} \geq \frac{\text{OPT}}{2}.$$

- If $|E| < k$, the contribution of any element $e \in V$ to a subset of $E$ is at most $\frac{\text{OPT}}{2k}$ which by submodularity of function $f$ shows that $f(e|E) \leq \frac{\text{OPT}}{2k}$. By summing up this inequality for all the elements $e$ in some optimal solution $O$, from submodularity we get that

$$f(O|E) \leq \frac{\text{OPT}}{2}.$$

Now, by monotonicity, it follows that $\text{OPT} \leq f(O|E) + f(E)$ and hence

$$f(E) \geq \frac{\text{OPT}}{2}.$$

Now consider the deletion of an element $e$. If $e \notin E$, then we do nothing since the current solution has the desired guarantees; we simply remove $e$ from all the sets in our hierarchy. Assume instead that an element $e_\ell \in E$ is deleted. Then we need to recompute the hierarchy beginning from $H_\ell$. Fortunately, this does not change the expected oracle-query complexity much as, intuitively, one needs to delete roughly $O(|H_\ell|)$ elements to delete a fixed element $e_\ell$ with large probability (since it is a randomly chosen element of $H_\ell$). Moreover, the time needed for this re-computation is $O(k \cdot |H_\ell|)$, which results in an amortized running time of $O(k)$. We introduce some additional optimizations.

**Improving the insertion algorithm**    We introduce two modifications to the previously described algorithm.

- **Capping the number of elements in $H$.** We ensure that the number of elements in $H_\ell$ is at most $2^{\log|V|-\ell}$, which also guarantees that the height of the hierarchy is at most $\log|V| + 1$. We achieve this by running more than one round of peeling. More precisely, when constructing $H_\ell$, we keep running rounds of peeling until either its size becomes below the desired threshold (i.e., $|H_\ell| \leq 2^{\log|V|-\ell}$), or we have selected $k$ elements (i.e., $|E| = k$). This does not hurt the running time nor the approximation guarantee.

- We maintain buffer sets $B_1, ..., B_T$, where $T = \log n + 1$ is the maximum possible height of the hierarchy. When an element is inserted, we add it to all the sets $B_\ell$. When, for any $\ell$, the sizes of $B_\ell$ and $H_\ell$ are equal we add the elements of $B_\ell$ to $H_\ell$ (and empty the set $B_\ell$). The main goal of this procedure is to handle updates lazily. During the execution of the algorithm, $B_\ell$ essentially represents those elements that we have not considered in the construction of $H_\ell$. Note that, as described before, the running time for constructing $H_\ell$ is at most $O(k|H_\ell|)$. This guarantees that the amortized running time for insertion is also $O(k)$. Also observe that we merge $B_T$ whenever its size is one. This enables us to maintain a good approximate solution at all times.[12]

**Improving the deletion algorithm**    Deletions are also handled in a lazy manner: we update our solution only when an $\epsilon$-fraction of elements in a set is deleted. In the next section we explain this idea in more details. Intuitively, we maintain a partitioned version of $H_\ell$ into sets $A_{i,\ell}$ consisting of elements with similar marginal contributions. When an $\epsilon$-fraction of elements in one set $A_{i,\ell}$ is deleted, we trigger a recomputation. Interestingly, same at the proofs presented before we can show that this significantly reduces the number of re-computations while giving a slightly weaker approximation guarantee, i.e., almost $1/2 - \epsilon$.