[Reviews · NeurIPS 2020]

Review 1

Summary and Contributions: This paper studies the problem of maximizing a monotone submodular function under a cardinality constraint in a fully dynamic setting that allows an arbitrary number of element insertions and deletions. This is the first paper showing guarantees (a 1/2- eps approximation) for this general dynamic setting with poly-log amortized update time. Previous results hold either for the more specialized setting of sliding windows or robustness which has an update time linear in the number of elements deleted. The authors also empirically evaluate their algorithm in dynamic settings with a large number of insertions and deletions.

Strengths: - Strong theoretical result: first algorithm to obtain guarantees for a general dynamic setting with an arbitrary number of insertions and deletions. - Interesting and non-trivial algorithm and analysis

Weaknesses: - The experiments do not use the algorithm with theoretical guarantees, but a variant of it. This is problematic because the baselines are other algorithms with theoretical guarantees that might also have variants with better empirical performance. For fairness, the experiments should use the algorithm analyzed in this paper.

Correctness: Overall, yes. One issue is with the algorithm used in the experiments, as discussed above.

Clarity: Yes, very well.

Relation to Prior Work: Yes, the related work section provides a clear comparison to related problems and settings.

Reproducibility: Yes

Additional Feedback:


Review 2

Summary and Contributions: This paper considers submodular optimization subject to a cardinality constraint in a dynamic setting, where a stream of insertions and deletions to the ground set occur. The paper proposes an algorithm that maintains a solution throughout the stream with a constant approximation ratio to the optimum, and only does amortized polylogarithm in n many oracle evaluations per insertion/deletion. While previous work has considered related problems with only insertions, the main new difficulty is handling deletions.

Strengths: This paper is the first to introduce the dynamic setting for submodular optimization. This setting is relevant to the submodularity community and would seem to have plenty of interesting applications. The algorithms and analyses appears to be novel and nontrivial.

Weaknesses: The description of the algorithm (sections 3 and 4) could use a lot of editing for clarity, since it's currently difficult to read. In addition, their results should be compared to related work in greater detail especially [FMZ19].

Correctness: I didn't find any issues.

Clarity: It could use significant improvement, but it is reasonably well written.

Relation to Prior Work: Additional detail could be added, as described in the weaknesses section.

Reproducibility: Yes

Additional Feedback: - How does this relate to online algorithms? - n and OPT are used in Section 3 though they are not defined. - Is this work related to the approach used by [1]? - It looks like there is no assumptions about a distribution on the insertions/deletions stream, but it could be interesting to look at that. [1] Kazemi, Ehsan, et al. "Submodular Streaming in All Its Glory: Tight Approximation, Minimum Memory and Low Adaptive Complexity." International Conference on Machine Learning. 2019.


Review 3

Summary and Contributions: This work studies dynamically maintaining a set of a fixed size that approximately maximizes a monotone submodular function when elements are arbitrarily inserted and deleted. This model is more general than the sliding window model and the deletion robust model. The authors proposed a randomized algorithm that has approximation ratio roughly 1/2 and polylogarithmic update time. Also the authors numerically compared the proposed algorithm with other baseline algorithm (though those baseline algorithms cannot handle the fully dynamic setting.)

Strengths: The fully dynamic setting is very natural and the obtained result is significant. Also the algorithm seems not to be easy to obtain, though I'm not very sure about this because I'm not an expert on dynamic algorithms. The empirical results is not very impressive in terms of the number of oracle calls and the solution quality, but it's fine because the fully dynamic setting is far more general than other models (the sliding window and deletion robust models) only in which other baseline methods work. Also the existence of empirical results show that the proposed algorithm is implementable.

Weaknesses: I couldn't find any.

Correctness: I couldn't find any apparent error, though I didn't check all the details.

Clarity: It's hard to get the idea of the correctness (especially about the approximation gurantee) of the proposed algorithm by reading Section 3.

Relation to Prior Work: Yes.

Reproducibility: Yes

Additional Feedback:


Review 4

Summary and Contributions: The authors design a fully dynamic data structure for maximizing monotone submodular functions. Given a sequence of insertion and deletion operations, their method can be used to return an $1/2-\epsilon$-approximate solution at any point while incurring an amortized polylogarithmic update time per operation. Previously there has been algorithm with similar guarantees for streaming sub-modular maximization which can only handle insertions. The main ideas allowing the authors to efficiently handle deletions are the following: - During the execution, multiple instances of the algorithm are maintained to provide multiple guesses for OPT. After each deletion, only those instances which are affected get updated. - Updates are done in a lazy manner; insertions and deletions get processed in batches to reduce the amortize update time. They also crucially use an algorithm from [1] which can be used to uniformly generate a subset of elements where the average contribution of all of its elements are ``large''. The approximation criteria as they mentioned is tight in the streaming (low memory) setting but has room to improve in their dynamic setting. 1. Matthew Fahrbach, Vahab S. Mirrokni, and Morteza Zadimoghaddam. Submodular maximization with nearly optimal approximation, adaptivity, and query complexity. In Proceedings of the Thirtieth Annual ACM-SIAM Symposium on Discrete Algorithms, SODA 2019, San Diego, California, USA, January 6-9, 2019, pages 255–273, 2019.

Strengths: Submodular maximizaiton is an extensively studied problem. Their result is theoretically and practically interesting. On the theoretical side, they offer the first type of data structures which can handle both insertion and deletions efficiently. Moreover, the setting studied here seems very natural in practice, and their algorithm can be of interest in many tasks.

Weaknesses: The description of the algorithm in the main body of paper was hard to follow. In particular, section 3 can be improved. Some parts of that was very difficult to me to understand before reading the supplementary material and section 4.

Correctness: I verified the details for the oracle complexity part, but didn't manage to completely read the correctness proof; the high level idea seemed in the right direction, though.

Clarity: Overall yes, but to understand some parts better you might need to skim the supplementary manuscript as well.

Relation to Prior Work: Yes.

Reproducibility: Yes

Additional Feedback: - The writing of section can be improved. In particular, I couldn't follow most of that in the first read, but it became much more readable after skimming through section 4. I think, presenting the high level idea behind each of the sets A,B, and S at the beginning of section 3 can make be helpful.

[Author Response · NeurIPS 2020]

We thank all the reviewers for their careful reading and valuable feedback. Below, we provide our responses to individual comments.

**Reviewer 1**:

• *The experiments do not use the algorithm with theoretical guarantees, but a variant of it.*

We apologize for not being precise enough in our explanation. Namely, the implemented algorithm also has theoretical guarantees, but the amortized update time is $\tilde{O}(k)$ as opposed to $O(\text{poly} \log n)$ achieved by our main approach. Both the main approach and the implementation achieve a $(1/2 - \epsilon)$-approximation. In Appendix E, we explain why this modified algorithm achieves these guarantees, and also why we believe that this simplified approach has good empirical performance. Notice that the analysis of our implementation is a simple version of the analysis of our main algorithm. We will make this statement formal in the camera ready version and add the following theorem:

**Theorem 1** *The implemented algorithm maintains a $(1/2 - \epsilon)$-approximate solution after each operation. The amortized expected number of oracle queries per update of this algorithm is $\tilde{O}(k)$.*

• *This is problematic because the baselines are other algorithms with theoretical guarantees that might also have variants with better empirical performance.*

We did take the utmost care to optimize the implementations of the baselines as well. For example, the Sieve-Streaming implementation we have used only recomputes its sub-sieves lazily as needed, which gives it a large boost in performance.

**Reviewer 2**:

Thanks for raising this point, we will clarify those aspects in the final version. In Appendix C we explain the subroutine (algorithm Peeling) of [FMZ19] used in our paper. We also state that the subroutine is proposed in [FMZ19]. All the other necessary ingredients, e.g. bucketing or the way we perform lazy evaluation, are novel ideas developed in this work. The work [FMZ19] addresses the different setting of adaptivity complexity, which has very little in common with the dynamic setting. We will add a detailed comparison between [FMZ19] and our submission.

• *How does this relate to online algorithms?*

The two settings are related but they study different objectives. In designing online algorithms one focuses on building a stable solution on the fly. Here instead we design an algorithm to *efficiently* compute a good solution at any point in time.

• *n and OPT are used in Section 3 though they are not defined.*

We will make additional passes on Sections 3 and 4. OPT is defined in Preliminaries, but we will recall its definition in Section 3.

• *Is this work related to the approach used by [1]?*

We think that the main contribution of the paper is to carefully handle a fully dynamic stream with addition and deletions. In particular, to handle deletions we have to introduce new ideas so we think that the work is only vaguely related to [1].

• *It looks like there is no assumptions about a distribution on the insertions/deletions stream, but it could be interesting to look at that.*

Indeed, our guarantees hold for an arbitrary distribution of insertions and deletions. It is a very interesting question whether one can obtain stronger guarantees when operations follow certain distributions, e.g. arrive in a random order. Thank you for pointing this out!

**Reviewer 3 and Reviewer 4**:

We will make additional passes over Sections 3 and 4 and improve their clarity. In particular, we will adopt R4's suggestion :"*I think, presenting the high level idea behind each of the sets A,B, and S at the beginning of section 3 can make be helpful.*"

[Meta-Review · NeurIPS 2020]

A strong, resounding accept.